# Comparison of Optical and Electrical Sensor Characteristics for Efficient Analysis of Attachment and Detachment of Aptamer

**DOI:** 10.3390/bios12110979

**Published:** 2022-11-07

**Authors:** Yejin Park, Thinh Viet Dang, Uiseok Jeong, Moon Il Kim, Jinsik Kim

**Affiliations:** 1Department of Biomedical Engineering, College of Life Science and Biotechnology, Dongguk University, Seoul 04620, Korea; 2Department of BioNano Technology, Gachon University, Seongnam 13120, Gyeonggi, Korea; 3SKhynix, Gyeongchung-daero 2091, Bubal-eup, Incheon-si 17336, Gyeonggi-do, Korea

**Keywords:** aptamer, graphene oxide, field-effect transistor, fluorescence quenching, cantilever-integrated waveguide, thrombin detection

## Abstract

Nucleic acid aptamer-based research has focused on achieving the highest performance for bioassays. However, there are limitations in evaluating the affinity for the target analytes in these nucleic acid aptamer-based bioassays. In this study, we mainly propose graphene oxide (GO)-based electrical and optical analyses to efficiently evaluate the affinity between an aptamer and its target. We found that an aptamer-coupled GO-based chip with an electrical resistance induced by a field-effect transistor, with aptamers as low as 100 pM, can detect the target, thrombin, at yields as low as 250 pM within five minutes. In the optical approach, the fluorescent dye-linked aptamer, as low as 100 nM, was efficiently used with GO, enabling the sensitive detection of thrombin at yields as low as 5 nM. The cantilever type of mechanical analysis also demonstrated the intuitive aptamer–thrombin reaction in the signal using dBm units. Finally, a comparison of electrical and optical sensors’ characteristics was introduced in the attachment and detachment of aptamer to propose an efficient analysis that can be utilized for various aptamer-based research fields.

## 1. Introduction

The demand for biosensors capable of providing point-of-care (POC) testing has recently increased as a result of the exponential spread of infective diseases and the need for patient self-testing. POC testing emphasizes the decrease in the size of the biosensor and simple steps for normal or rapid detection. The biosensors can be classified based on the type of transducing method, such as piezoelectric, electrical, mechanical, optical, or chemical biosensors [1,2,3]. Among them, electrical and optical biosensors have been in the spotlight lately because these sensors can accomplish the demand for POC and self-testing compared with traditional biosensors. Furthermore, numerous researchers are trying to improve sensing performance by utilizing various nanomaterials such as graphene as transducing materials for biosensors [4]. Biosensors frequently involve bioreceptors, such as a protein-based antibody or a nucleic acid-based aptamer, to detect target analytes that have a pathological significance. These bioreceptors have played an important role in achieving selectivity and sensitivity in pathogen detection. Aptamers, mainly single-stranded nucleic acids, have been highlighted as a substitute for antibodies because of their various advantages, such as their physical and chemical stability, cost-effectiveness, and ability to easily be used for various targets [5]. Moreover, because the need for biosensors for POC testing is increasing, the need for a stable and inexpensive receptor to enhance the sensor’s performance should be emphasized. Thus, an aptamer-based strategy may address these needs.

Aptamers can be produced with the systematic evolution of ligands by exponential enrichment (SELEX). Preexisting SELEX, particularly based on a target-immobilization method, can change the structure of the target molecules when the target molecules have a relatively small size or molecular weight, and eventually, it may lead to undesired effects and limits the selection of aptamers in many practical applications [6]. Therefore, SELEX needs a monitoring method in its intermediate stage during the cycles, and there are lots of methods, such Nuclear Magnetic Resonance and real-time PCR [7,8,9]. However, those approaches are complicated and time-consuming. Thus, methods for direct quality control are still needed. From that point of view, the GO-SELEX can be an appropriate approach for quality control during the SELEX [10,11]. However, the GO has a wide 2D plane with high adsorption properties that can react directly to the DNA and RNA without chemical linkers [12,13]. However, the qualification methods for that process are still challenging, even if the GO-SELEX is a great approach, because there is a potential to absorb even a well-fabricated aptamer.

However, it is a well-known fact that GO is a universal substance that can be used not only for producing aptamers but also for developing biosensors based on aptamers because GO has various characteristics including high mechanical strength and easy fabrication [14]. Moreover, the fact that GO can elevate its electrical conductivity through a reduction process and has a fluorescence-quenching effect also enables GO to be utilized in various types of biosensors with aptamers, including electrical and optical sensors [15,16,17]. To be specific, among the various aptamer-based GO-sensor studies, the most actively studied field is a sensor targeting thrombin [18,19,20,21]. Thrombin is a protease, and activated thrombin is involved in blood coagulation by converting soluble fibrinogen into insoluble fibrin [22,23]. In blood coagulation, damage to the inhibitory function of thrombin can lead to an abnormal increase in the thrombin concentration in the body and an overcoagulation of the blood. This overcoagulation results in severe consequences, such as the narrowing of blood vessels. Hemostatic diseases, such as blood coagulation disorders, can be diagnosed by measuring the thrombin concentration, suggesting that thrombin can act as a biomarker for these diseases. Thrombin is a substance that is always present in the body, and its concentration may vary with the severity of the disease, therefore a sensor capable of measuring the concentration of thrombin with high accuracy is essential for the diagnosis of diseases. Despite the pathological significance of thrombin and the widespread use of aptamers, comparable research among various analysis methods with regulated variables, such as using the same sensing substrate, is still lacking.

In this study, we conclusively describe two potent analyses based on electrical and optical strategies to efficiently evaluate the affinity of aptamers toward their target for comparative analysis and applying them to the detection of a target within a broad concentration range. The experiment in this study is based on an aptamer’s adsorption and desorption when it reacts to the target, thrombin. In this study, the mechanical method was also conducted, and it shows the obvious changes in accordance with the attachment and detachment of the aptamer. However, the result of the mechanical measurement was utilized only to help understand the intuitive reaction of the aptamer because of the insufficient characteristics of POC such as low disposability, huge size, etc. Through electrical and optical analysis, the affinity of an aptamer toward the target can be evaluated and verified. The feasibility of this system is also evaluated in the detection of thrombin. The electrical measurement method enabled us to quantify the obtained outputs from the attachment and detachment of a thrombin-binding aptamer (TBA) in the presence of the thrombin. The optical system enabled the reliable determination of both the TBA and thrombin; however, a fluorescent dye must be conjugated on the TBA. These analytical systems can be used to evaluate the affinity of aptamers toward their target, which may help to detect analytes with a pathological significance when the aptamers are utilized as bioprobes for biosensors. The overall concept of this paper has shown in Figure 1. The details of atomic force microscopy images in the Figure 1 was described in Appendix A.

## 2. Materials and Methods

### 2.1. Chemicals and Reagents

Hydrogen iodide (HI), N-(3-dimethylaminopropyl)-N′-ethylcarbodiimide, N-hydroxysuccinimide, Trizma base, glucose oxidase (GOx), trypsin, lysozyme, ficin, and thrombin from bovine plasma were purchased from Sigma-Aldrich Inc. (Merck KGaA, Germany). Goat anti-mouse immunoglobulin G (IgG) was purchased from Boreda Biotech (Seongnam, Korea). Acetone 99.8%, isopropyl alcohol (IPA) 99.9%, and deionized water (DI water) were purchased from Daejung Chemical and Metals Co., Ltd. (Daejung, Korea). Phosphate-buffered saline (PBS) GibcoTM with pH 7.4 was purchased from Thermo Fisher scientific (Thermo Fisher Inc., Walham, MA, USA). A GO dispersion in water was purchased from Graphene Supermarket (Graphene Laboratories Inc., Calverton, NY, USA). DNA aptamers for thrombin (TBA) with and without a fluorescein phosphoramidite (FAM) fluorophore were purchased from Integrated DNA Technologies (Coralville, IA, USA) and the amine-modified TBA was designed by Bioneer Inc. (Bionner, Korea). The sequence information for the TBA, TBA-FAM, and TBA-amine are 5′-GGTTGGTGTGGTTGG-3′, 5′-FAM-GGTTGGTGTGGTTGG-3′, and 5′-C6 Amine-GGTTGGTGTGGTTGG-3′, respectively. 

### 2.2. Mechanical Measurement

#### 2.2.1. Fabrication of a Cantilever-Type Mechanical Sensor

A cantilever-type mechanical sensor was utilized to test the interaction between the aptamer and the thrombin based on the its mass. The mechanical-based sensor can generally provide intuitive results according to the presence and absence of the target materials. The device was integrated with a Silicon (Si) waveguide to measure the deflection of the cantilever via the coupling between high-index-contrast Si waveguides and the lensed fibers. E-beam lithography was achieved with e-beam resists (ZEP-520) for the waveguide shapes [24]. The strip waveguide was patterned using reactive ion etching (RIE) to form an etch depth of 250 nm. After the e-beam resist was stripped, a passivation layer of 3.19 μm-thick silicon dioxide (SiO_2_) was deposited via plasma-enhanced chemical vapor deposition at 250 °C and 900 mTorr. The Si waveguides were fabricated on 200 nm silicon on an insulator wafer via e-beam lithography and RIE methods and were buried in the SiO_2_ cladding.

#### 2.2.2. Analysis of the Mechanical Sensor’s Output Signal

The mechanical measurement method was based on the degree of cantilever bending when the aptamer or its target material was absorbed onto its surface (Figure 2a). About 0.38 dB/mm propagation loss and 3.4 dB/facet coupling loss of the rib waveguide were measured using an optical signal analyzer (ANDO, AQ6319, bandwidth: 600~1700 nm, wavelength accuracy: ±10 pm, resolution: 0.02 nm, resolution accuracy: ±6%) at a 1550 nm wavelength. Continuous wave light from a tunable laser (SANTEC, 81989A) was coupled to the device using a 5 μm-diameter lensed fiber, while a polarization controller controlled the laser source. wavelength of 1550 nm was used as an input signal that passed through the laser diode to the lensed fiber. The light intensity was altered according to the degree of cantilever distortion induced by the mass of the molecules on its waveguide surface. The data are described using dBm units. The reaction mechanism between the molecules was verified through the change in optical intensities according to the reaction of the molecules.

### 2.3. Electrical Measurement 

#### 2.3.1. Fabrication of the Field-Effect Transistor (FET) Sensor

The field-effect transistor (FET) sensors were fabricated using microelectromechanical system techniques. First, a GO solution was deposited on a cleaned 4 inch SiO_2_ with meniscus dragging. The deposited GO was reduced with HI and etched with RIE for patterning. The dimension of the GO was designed based on the previous study of this group [25,26,27]. A photoresist (PR) AZ GXR 601 was spin-coated onto the wafer, and the coated wafer was exposed to ultraviolet light. After developing the PR, the Chromium/Gold (Cr/Au) layer was deposited to a thickness of 100/1000 Å using e-beam evaporation at 0.1 nm/s. Finally, the electrode layer was formed via a lift-off process, followed by rinsing with DI water and IPA (Appendix A). 

#### 2.3.2. Analysis of the Electrical Method

In the electrical measurement method, GO was adopted as a transducer material in the FET biosensor (b). The experimental protocol of the FET sensor-based electrical method is described here. First, the aptamer was absorbed on the GO sensing zone of the bare FET sensor for 30 min. Afterward, thrombin was added and washed with wash buffer, PBS, and DI water. The electrical current from the source to the drain part of the FET sensor was measured and recorded while voltage was applied to the electrode with a source meter (Keithley Smus–2401, A Tektronix Company, North Billerica, MA, USA). The applied voltage range was 0–5 V, and the voltage was changed at 0.05-V intervals. The electrical characteristics of the sensor alternated depending on the electrical charge of the molecules in the sensing zone because the sensor has a characteristic that its electrical conductivity increases when the molecule is positively charged and decreases when it is negatively charged. Accordingly, the results were analyzed with the distribution of the resistance change rate. 

### 2.4. Optical Measurement 

#### Analysis of the Optical Method: Fluorescence Quenching and Restoration

The optical measurement method involved fluorescently tagged TBA followed by its reaction with the GO and thrombin, thus inducing fluorescence quenching and restoration, respectively (c). The fluorescence quenching of the TBA-FAM via the interaction with GO was evaluated based on its variance of fluorescence intensity. First, aqueous dispersions of GO (100 μL, 0–1.5 mg/mL) were incubated with the TBA-FAM (100 μL, 1 μM) in a Tris-HCl buffer (800 μL, 50 mM, pH 7.4). The mixed solution was then incubated at room temperature for 30 min. The fluorescence intensity was measured using a microplate reader (Synergy H1, BioTek, VT, USA) via excitation and emission wavelength at 440 nm and 520 nm, respectively. The TBA was thought to be adsorbed on the surface of the GO via non-covalent π–π stacking interactions, yielding efficient quenching through fluorescence resonance energy transfer (FRET) between the GO and the FAM conjugated on the aptamer [28].

The thrombin levels were quantified by measuring the restoration of the fluorescence using the TBA-FAM/GO-based strategy. Briefly, various concentrations (0–10 μM) of thrombin were added to the mixtures of GO and TBA-FAM, followed by further incubation for 15 min. Then, the fluorescence intensity was measured as described above or via emission scanning. The selectivity for detecting thrombin was evaluated by adding other interfering protein substances such as GOx, IgG, trypsin, lysozyme, or ficin at 100 μM.

## 3. Results

### 3.1. Measurement of Aptamer Adsorption Using Various Types of Sensors

Non-covalent π–π stacking interactions primarily induce the reactivity of TBA to substrate materials and are the basis of each detection method. A schematic diagram of the experimental principles is shown in a–c. 

#### 3.1.1. Detection of Aptamer Adsorption via the Mechanical Method

A mechanical cantilever sensor was utilized to measure the adsorption of aptamers based on their mass, intuitively. The mechanical method confirmed the TBA adsorption using changes in optical intensity through a fiber based on the change in mass due to adsorption. The initial peak value of the cantilever coupler’s optical intensity was −5.351 dBm at around 1550 nm. The optical intensity changed to approximately −7.185 dBm when the TBA was absorbed onto the surface of the cantilever (d). This difference after the reaction of TBA can be rewritten as a unit of percentage according to Equation (1).
**Y = 10^0.1×A^**(1)
In the equation, A refers to the difference values between the signals before and after adsorption of the TBA, and Y refers to the conversion of that difference from a dB scale to a unit-of-percentage ratio. According to Equation (1), the signal decreased by about 152.546% after the TBA adsorption compared with the chip before adsorption, and the signal difference between the two steps was about −1.834 dB. TBA with a molecular weight of about 5 kDa was utilized in the experiment at a concentration of 100 nM [29].

#### 3.1.2. Detection of Aptamer Adsorption Using the Electrical Method 

Using the electrical method, changes in the electrical characteristics of the FET sensor, such as changes in the electrical current between the source and the drain, were verified to correlate with the concentrations of TBA. The electrical current was measured by applying voltages through the source port and was converted to an electrical resistance change rate (Appendix A). The concentration of the TBA used ranged from 100 pM–20 nM. As expected, the resistance change rate of the electrical sensor gradually increased from 2.802% to 10.422% (e). In a linear calibration, 100 pM was calculated to be the limit of detection (LOD) (R^2^ = 0.7740). These results show that a very low concentration of aptamer was detected using the electrical method.

#### 3.1.3. Detection of Aptamer Adsorption Using the Optical Method

We also analyzed the aptamer adsorption on the GO substrate via fluorescence quenching using the TBA-FAM. Various concentrations of GO and TBA-FAM were incubated together to optimize the quenching efficiency via π–π interaction-driven FRET with GO. It was found that 0.03 mg/mL of GO and 0.1 μM of TBA were optimal for inducing fluorescence quenching and were used for the following experiments. The quenching efficiency (QE) was calculated as follows:**QE (%) = (F_0_ − F_1_)/F_0_ × 100**(2)
F_0_ and F_1_ represent the fluorescence intensity of the TBA-FAM without and with GO, respectively. Thus, to evaluate the effects of GO on its quenching ability, a constant concentration of TBA-FAM was used, while the concentration of GO varied. Different quantities of TBA-FAM were used and varied in both F_0_ and F_1_ values, and F_0_/F_1_ ratios were generated to optimize the concentration of the aptamer. As the concentration of aptamers increased to 0.1 μM, an inverse relationship between F_0_ and F_1_ was observed, resulting in an enhanced quenching capacity (f). However, when the concentrations were raised beyond 0.1 μM, both F_0_ and F_1_ values increased comparatively, leading to quenching capacity saturation as the ratio of F_0_ and F_1_ was unchanged. 

### 3.2. Detection of Thrombin with the TBA-Absorbed Substrate 

The variances of the mechanical, electric, and optical signals after the thrombin addition on the TBA-absorbed sensing substrate of each method were analyzed. 

#### 3.2.1. Detection of Thrombin with the Mechanical Method

The cantilever coupler had a reconstructed value after the addition of thrombin to the TBA-absorbed substrate. The intensity after the addition of the thrombin was −5.952 dBm, showing a difference of about 1.233 dB from the initial value of TBA only (Figure 3a). This result is based on a consequence of the desorption of the TBA due to the deformation of its structure when it reacts with thrombin. Because the TBA was physically absorbed onto the cantilever’s surface, the deformation of the TBA could make it detached from the surface. The following difference between the two signals was calculated by Equation (1), and as a result, the quantity of the signal dropped because of the aptamer response and was recovered at a yield of 132.831%.

#### 3.2.2. Detection of Thrombin with Electric Method

The electrical resistance change rate after the addition of thrombin had a negative number when it was calculated by comparing it with the electrical resistance of the TBA without thrombin (Figure 3b). The concentrations of the TBA and thrombin used in this experiment were 200 nM and 250 pM, respectively. The absolute value of the electrical resistance change rate after the addition of thrombin decreased as the thrombin reaction time increased and had a maximum absolute value at the incubation time of five minutes. We verified that the desorption reaction due to the TBA–thrombin interactions was performed within five minutes, and the reactivity of the biosensor decreased as the reaction time increased because of the error in the signals from non-specific substances.

#### 3.2.3. Detection of Thrombin with the Optical Method

In the optical measurement method, the TBA detached from the GO when thrombin was added, resulting in the restoration of the fluorescence from the FAM conjugated on the TBA [30]. The selectivity in detecting the thrombin was explored with interfering protein substances, such as GOx, IgG, trypsin, lysozyme, and ficin, at ten-fold higher concentrations than the thrombin. These results show that solely thrombin induced restoration of the fluorescence intensity, whereas the other interfering substances did not induce any meaningful signal (Figure 3c), confirming the selectivity of the TBA-FAM/GO-based strategy to optically detect the target. As the concentrations of thrombin increased, the respective fluorescence intensities increased and fit the linear calibration plot in a range of 0.0625–100 μM with a high R^2^ value greater than 0.99 (Figure 3d). The LOD for the thrombin was calculated to be approximately 5.0 nM, thrice the standard deviation of the signal from the blank divided by the slope of the calibration curve. 

### 3.3. Comparison between the Electrical and Optical Methods

This study used electrical and optical analysis methods to evaluate the efficiency and utility of the affinity between the aptamer and its target. The full-scale input (FSI) and full-scale output (FSO) derived from the reactivity of each concentration of the target, TBA, and thrombin were used as parameters in the comparison analysis. The FSI and FSO indicate the reliable range of results calculated from each method. The R^2^ value in Figure 4 can be interpreted as an indicator of the linearity in the sensing methods. The sensitivity of each sensing method was calculated on the basis of the slope value of the FSO and FSI. 

This study indicates that electrical analysis could be an appropriate method for the verification of extremely low concentrations of aptamer in the fM to pM range and has a high sensitivity to validate the presence of the aptamer (Figure 4a). However, the electrical analysis might be unsuitable for use, because the reactivity is confirmed within a very short time resulting from the absolute value of the resistance change rate decreasing as the thrombin reaction time increases. In addition, unlike aptamer detection, it was very difficult to verify thrombin at a low concentration (Figure 4b). These results show that the electrical analysis used in this study is more useful for analyzing a single substance than multiple ones, such as detecting the aptamer and target together. 

The optical analysis method had the advantage of confirming the aptamer’s concentration and having a high efficiency in detecting thrombin. These results prove the potential of the optical strategy for the selective and sensitive determination of thrombin and the binding affinity between thrombin and TBA; however, it was difficult to analyze the reactivity of thrombin at a relatively low concentration. Therefore, the optical and electric methods could be simultaneously utilized to analyze the affinity between the aptamer and its target.

## 4. Conclusions

In this study, mechanical, electrical, and optical methods were tested and evaluated for efficacy to analyze the affinity between TBA and its target, thrombin. The cantilever-based mechanical method was efficient in intuitively detecting the presence of the aptamer on the substrate. The FET-based electrical method was efficient in detecting the TBA at extremely low concentrations down to the picomolar level; however, the quantitative detection of thrombin was not efficient, possibly due to interfering molecules. The optical method based on fluorescence quenching and restoration enabled the successful detection of the TBA and thrombin, although the detection limit was a limitation. These analysis methods can be cooperatively utilized to screen appropriate aptamers in GO-SELEX and as analytical techniques in aptamer-mediated biosensing. Among aptamer sensors, those that use thrombin as a target material are being actively researched [31,32,33]. However, there are some limitations in using it as a monitoring method for GO-SELEX, as studies using graphene-based nanomaterials utilize chemical immobilization methods such as EDC/NHS [34,35,36]. Moreover, even if there are papers based on adsorption, it is difficult to find papers using the same material in sensing, such as graphene oxide, so it is difficult to assess and select an appropriate sensing method depending on the purpose [37,38]. Therefore, we think that the results of this study possibly propose a reference point that can be used alone or in combination with a monitoring method or a sensing method in accordance with the various purposes of various aptamer and thrombin applications.

## Figures and Tables

**Figure 1 biosensors-12-00979-f001:**
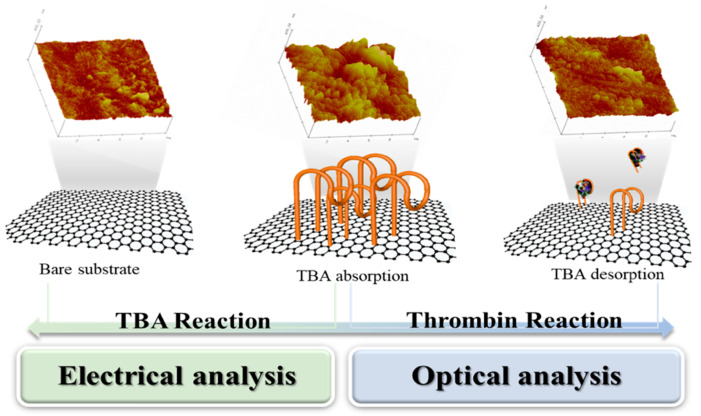
The schematic view of main theme which is comparison between electrical and optical analysis for attachiment of aptamer and detachment of aptamer by thrombin. The attachment and detachment of aptamer was verified with atomic force microscopy images.

**Figure 2 biosensors-12-00979-f002:**
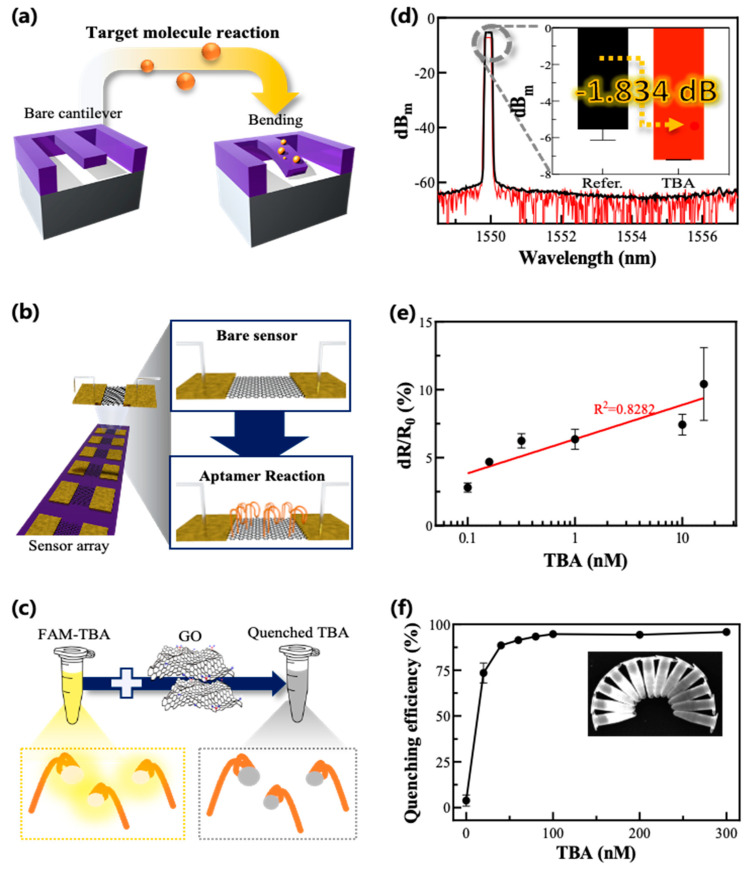
Experimental principles and schematic diagrams are based on each measurement method and experimental results after TBA absorption on the sensing substrate. (**a**,**d**) shows an illustration and a representative graph showing the mechanical method. (**b**,**e**) shows an illustration and representative graph showing the electrical method. (**c**,**f**) shows that the standard deviation (SD) value after TBA reaction in (**d**) is about 0.018. All experiments except for the highest concentration of TBA in (**e**) have SD values under 0.74. The highest concentration condition, 20 nM TBA, has an SD value of 2.681. In (**f**), the SD values of two points under the 20 nM TBA concentration were 3.081 and 5.493, in ascending order.

**Figure 3 biosensors-12-00979-f003:**
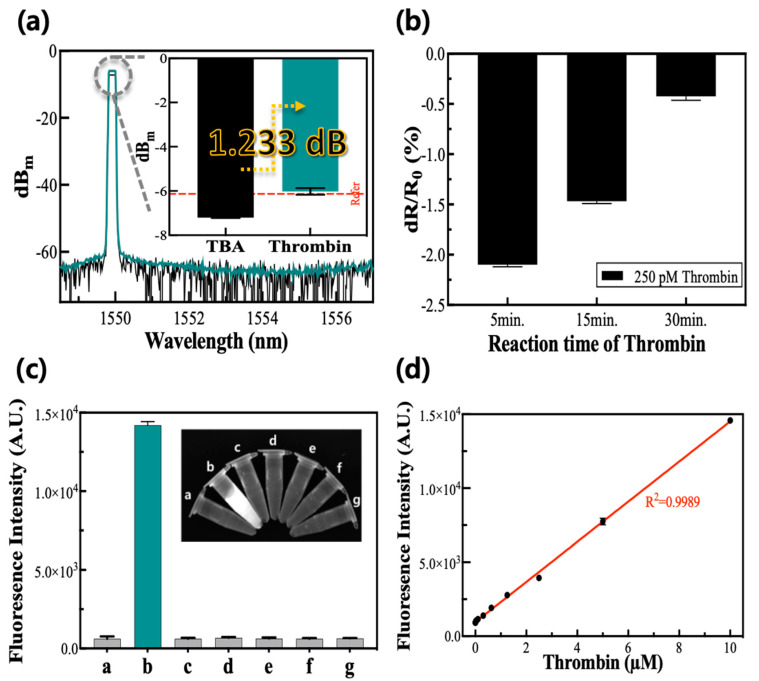
Detection of thrombin via mechanical, electric, and optical methods. (**a**) Mechanical analysis before and after the addition of thrombin on the aptamer that adhere to substrate. (**b**) Electrical analysis resistance change rate according to the reaction time of thrombin. (**c**) Optical analysis for the selectivity of the TBA-FAM/GO-based system. The letters (a–g) of the graph and inset picture represent the blank, thrombin, GOx, IgG, trypsin, lysozyme, and ficin. (**d**) The corresponding calibration plot to (**c**) shows the fluorescence intensity versus thrombin concentration. The SD value after thrombin reaction in (**a**) was about 0.152. In (**b**), the SD values increased over time from 0.020 to 0.038. The coefficient of variation of the experimental results in (**d**) has an average value of 0.028 (±0.016).

**Figure 4 biosensors-12-00979-f004:**
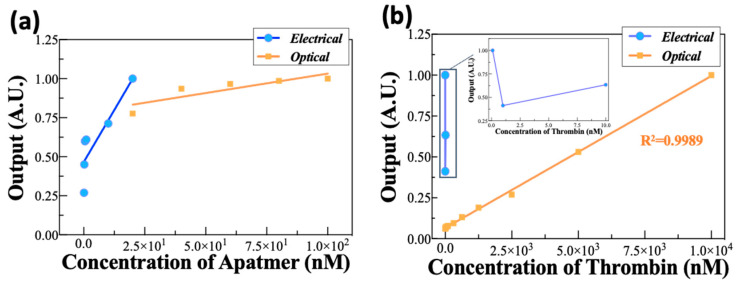
Comparison of the normalized experimental data using electrical and optical analysis. (**a**,**b**) were normalized according to the concentrations of aptamer and thrombin, respectively.

## Data Availability

The data presented in this study are available from the corresponding authors upon request.

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
