# Peer review of "Comparison of Optical and Electrical Sensor Characteristics for Efficient Analysis of Attachment and Detachment of Aptamer"

_biosensors, 2022, doi:10.3390/bios12110979_

Round 1

Reviewer 1 Report

This article provides three interesting methods to evaluate the affinity between the aptamer and its target including mechanical, electrical, and optical measurements. Nevertheless, a few issues are needed to be clarified.

1.     In the text, mechanical, electrical, and optical methods were presented to analyze aptamer absorption and affinity between TBA and the target. Why was the mechanical method not discussed in the abstract and section 3.3?

2.     The format of figure 3 is inconsistent with figure 1 and figure 2.

3.     It is described that the method developed in the paper can be used during GO-SELEX. More discussion or experiment data about the connection between the analysis methods and GO-SELEX should be added.

4.     Please check the formatting of the manuscript. There are several cases when blank space is not correctly used and font size is wrong.

5.     There are several grammatical errors.

Author Response

Reviewer 1

  1.    In the text, mechanical, electrical, and optical methods were presented to analyze aptamer absorption and affinity between TBA and the target. Why was the mechanical method not discussed in the abstract and section 3.3?

(Answer)

First of all, Thank you for your sharp question about structure of our manuscript.

As you pointed, the mechanical method was just shown at the figure 1 (a) and figure 2 (a). Because of issue for fabrication by COVID-19, we couldn’t have sufficient number of mechanical sensors. For the discussion in the abstract and section 3.3, we thought that mechanical sensors should be utilized according to concentration of TBA or thrombin as electrical and optical sensing.

However, the mechanical sensor can detect the attachment and detachment of TBA through changes of mass. So, we introduced the main body about the mechanical sensor to show application of various type of sensors for same biomolecules. And the mechanical sensors have meaning as a pre-test before electrical and optical sensing.

So, we also agree and understand that what is the main point of this question, we added the sentence below at the part of “2.2 mechanical measurement” to reduce the confusing of reader.

“The mechanical sensors can immediately detect the changes of mass according to attachment and detachment of aptamer. So, in this paper, the mechanical sensor was utilized as a pilot-test before electrical and optical sensing.

  1.    The format of figure 3 is inconsistent with figure 1 and figure 2.

(Answer)

 I apologize the inconvenience of difference in formatting. As follow your comment, we fixed the figures to have same format. Thank you for the comment to improve our manuscript.

  1.    It is described that the method developed in the paper can be used during GO-SELEX. More discussion or experiment data about the connection between the analysis methods and GO-SELEX should be added.

(Answer)

 First of all, we really apologize about the confusion of GO-SELEX. Originally, we tried to emphasize GO-SELEX as possible applications of sensors, because of the common materials among sensors and GO-SELEX, GO. Even though, we can describe the absorption and desorption mechanism between GO, Aptamer, and target material and based on this, we think that we can suggest those method for monitoring method in GO-SELEX.

And we fully agree with that current manuscript, especially introduction part, can make wondering the direct linkage with GO-SELEX. Thus, we edited that part to reduce ambiguity in research’s purpose and conclusion.

[before]

Aptamers can be produced by the systematic evolution of ligands by exponential enrichment (SELEX). Preexisting SELEX, particularly based on a target-immobilization method, can change the structure of the target molecule when the target molecules have relatively small size or molecular weight and eventually it may lead to undesired effects and limits the selection of aptamers in many practical applications [6]. Moreover, since a faulty aptamer with a low affinity can only be identified in the final screening step of SELEX, a delay in the experiment is inevitable. To overcome this limitation, non-immobilized target-based SELEX methods, such as AuNPs-SELEX and Graphene oxide (GO)-SELEX, were developed, which may offer better options for small molecule targets [7, 8, 9]. Especially, GO-SELEX is highly advantageous due to the various characteristics of GO, including high mechanical strength and easy fabrication [10]. Moreover, the GO-SELEX can be an appropriate approach in terms of interim verifying method for quality control during the SELEX while it still remains as a limitation to verify the state of aptamer during its intermediate stage in the SELEX although the flow of change to new method using non-immobilization-based SELEX for excavating the aptamers. Because it is much easier to check the generation and quality of aptamer during the intermediate stage of SELEX since the GO has a wide 2D plane with high absorption property that can react directly to the aptamer and target without chemical linkers [11, 12]. Consequently, GO is not only a versatile material for producing aptamers and developing biosensors but also required to develop a potent process for convenient, sensitive and selective evaluation of the affinity between aptamers and targets.

Therefore, we described two potent analyses based on electrical and optical strategies to efficiently evaluate the affinity of aptamers toward their target for comparative analysis in this study. GO was utilized as a substrate material because it can elevate its electrical property through a reduction process, has a fluorescence quenching property, and has a high-binding property with ssDNA-type aptamers.

[after]

Aptamers can be produced by the systematic evolution of ligands by exponential enrichment (SELEX). Preexisting SELEX, particularly based on a target-immobilization method, can change the structure of the target molecule when the target molecules have relatively small size or molecular weight and eventually it may lead to undesired effects and limits the selection of aptamers in many practical applications [6]. Therefore, SELEX needs to monitoring method in its intermediate stage during the cycles and there are lots of method such Nuclear Magnetic Resonance and real-time PCR [7, 8, 9]. However, those approaches are complicated and time-consuming, besides they can be used only for verifying method in the qualification. Thus, methods for quality control directly still need. In that point of view, the GO-SELEX can be an appropriate approach for quality control during the SELEX [10, 11]. However, the GO has a wide 2D plane with high absorption property that can react directly to the DNA and RNA without chemical linkers [12, 13]. Thus, the qualification methods are still challenging even if the GO-SELEX is one of great approach because it has a potential to absorb well-fabricated aptamer.

Even though, it is a well-known fact that GO is a universal substance that can be used not only for producing aptamers but also developing biosensors based on aptamer be-cause GO has various characteristics, including high mechanical strength and easy fabrication [14]. Based on these properties, there are lot of GO-biosensing method utilizing aptamer and the fact that GO can elevate its electrical property through a reduction process and has a fluorescence quenching property also can make various types of GO based biosensor with aptamer [15, 16, 17]. Among the various aptamer-based GO-sensor studies, the most actively studied is a sensor targeting thrombin [18].

  1. Please check the formatting of the manuscript. There are several cases when blank space is not correctly used and font size is wrong.

(Answer)

Thank you for the comment. We corrected the typo, blank or etc. what you told.

  1.    There are several grammatical errors.

(Answer)

Overall, we corrected the manuscript in authorized company to improve the English.

Reviewer 2 Report

The manuscript ‘Highly efficient analysis for the affinity between the aptamer 3 and target thrombin via electrical and optical methods’ by Park et al. describes aptasensors for a popular target, thrombin. This field is highly elaborated; dozens of biosensors have been proposed, including those with the aptamers and GO. The presented data have a limited novelty, as well as the LoDs are not the best. The reported aptasensors achieved LoD as low as 0.05 pM. The exhaustive review on the different aptasensors was provided by Sun et al. (Sun, H.; Wang, N.; Zhang, L.; Meng, H.; Li, Z. Aptamer-Based Sensors for Thrombin Detection Application. Chemosensors 2022, 10, 255. https://doi.org/10.3390/chemosensors10070255). Here the authors refer to GO-SELEX, but the manuscript does not contain any interaction between the proposed aptasensor and SELEX. So, the lack of the novelty is a crucial drawback of the work. One more point is a reference to PoC devices. The sensors were not tested with blood plasma, and their applicability to real still unclear. My last comment is about the analytical characteristics of the sensor. The linear correlation cannot be provided for the data with R2=0.82. The SD for the figure 1d is necessary to estimate the reproducibility of the signal.

Author Response

Reviewer 2

  1. The manuscript ‘Highly efficient analysis for the affinity between the aptamer 3 and target thrombin via electrical and optical methods’ by Park et al. describes aptasensors for a popular target, thrombin. This field is highly elaborated; dozens of biosensors have been proposed, including those with the aptamers and GO. The presented data have a limited novelty, as well as the LoDs are not the best. The reported aptasensors achieved LoD as low as 0.05 pM. The exhaustive review on the different aptasensors was provided by Sun et al. (Sun, H.; Wang, N.; Zhang, L.; Meng, H.; Li, Z. Aptamer-Based Sensors for Thrombin Detection Application. Chemosensors 2022, 10, 255. https://doi.org/10.3390/chemosensors10070255).

(Answer)

 First of all, thank you for the sharp points.

Our paper has novelty about comparing the characteristic of sensor’s response for the same aptamer and thrombin interaction; the attachment of aptamer and detachment of aptamer by thrombin which can be the model of aptamer fabrication in GO-SELEX. And there is obvious advantage and disadvantage according to the types of sensors. In our opinion, the comparison among the sensors according to the transducing method has novel scientific information, more than improvement of limit of detection. So, please, reconsider our paper through the point.

Furthermore, your commented paper is review paper which is researched with different target of aptamer and thrombin. So, it is difficult to say that is an exact comparison in a situation with variable-controlling. On the other hand, the paper also emphasizes the lack of a method to measure the concentration of thrombin. However, in our paper, it was shown that the amount of thrombin responding to the aptamer could be measured using through three kinds of sensors. Considering that, I think our paper is sufficiently valuable.

  1. Here the authors refer to GO-SELEX, but the manuscript does not contain any interaction between the proposed aptasensor and SELEX. So, the lack of the novelty is a crucial drawback of the work.

(Answer)

 We explain our manuscript’s novelty at answer for the 1st question, above. And we really apologize about the confusion of GO-SELEX. Originally, we tried to emphasize GO-SELEX as possible applications of sensors, because of the common materials among sensors and GO-SELEX, GO. Even though, we can describe the absorption and desorption mechanism between GO, Aptamer, and target material and based on this, we think that we can suggest those method for monitoring method in GO-SELEX.

And we fully agree with that current manuscript, especially introduction part, can make wondering the direct linkage with GO-SELEX. Thus, we edited that part to reduce ambiguity in research’s purpose and conclusion.

[before]

Aptamers can be produced by the systematic evolution of ligands by exponential enrichment (SELEX). Preexisting SELEX, particularly based on a target-immobilization method, can change the structure of the target molecule when the target molecules have relatively small size or molecular weight and eventually it may lead to undesired effects and limits the selection of aptamers in many practical applications [6]. Moreover, since a faulty aptamer with a low affinity can only be identified in the final screening step of SELEX, a delay in the experiment is inevitable. To overcome this limitation, non-immobilized target-based SELEX methods, such as AuNPs-SELEX and Graphene oxide (GO)-SELEX, were developed, which may offer better options for small molecule targets [7, 8, 9]. Especially, GO-SELEX is highly advantageous due to the various characteristics of GO, including high mechanical strength and easy fabrication [10]. Moreover, the GO-SELEX can be an appropriate approach in terms of interim verifying method for quality control during the SELEX while it still remains as a limitation to verify the state of aptamer during its intermediate stage in the SELEX although the flow of change to new method using non-immobilization-based SELEX for excavating the aptamers. Because it is much easier to check the generation and quality of aptamer during the intermediate stage of SELEX since the GO has a wide 2D plane with high absorption property that can react directly to the aptamer and target without chemical linkers [11, 12]. Consequently, GO is not only a versatile material for producing aptamers and developing biosensors but also required to develop a potent process for convenient, sensitive and selective evaluation of the affinity between aptamers and targets.

Therefore, we described two potent analyses based on electrical and optical strategies to efficiently evaluate the affinity of aptamers toward their target for comparative analysis in this study. GO was utilized as a substrate material because it can elevate its electrical property through a reduction process, has a fluorescence quenching property, and has a high-binding property with ssDNA-type aptamers.

[after]

Aptamers can be produced by the systematic evolution of ligands by exponential enrichment (SELEX). Preexisting SELEX, particularly based on a target-immobilization method, can change the structure of the target molecule when the target molecules have relatively small size or molecular weight and eventually it may lead to undesired effects and limits the selection of aptamers in many practical applications [6]. Therefore, SELEX needs to monitoring method in its intermediate stage during the cycles and there are lots of method such Nuclear Magnetic Resonance and real-time PCR [7, 8, 9]. However, those approaches are complicated and time-consuming, besides they can be used only for verifying method in the qualification. Thus, methods for quality control directly still need. In that point of view, the GO-SELEX can be an appropriate approach for quality control during the SELEX [10, 11]. However, the GO has a wide 2D plane with high absorption property that can react directly to the DNA and RNA without chemical linkers [12, 13]. Thus, the qualification methods are still challenging even if the GO-SELEX is one of great approach because it has a potential to absorb well-fabricated aptamer.

Even though, it is a well-known fact that GO is a universal substance that can be used not only for producing aptamers but also developing biosensors based on aptamer be-cause GO has various characteristics, including high mechanical strength and easy fabrication [14]. Based on these properties, there are lot of GO-biosensing method utilizing aptamer and the fact that GO can elevate its electrical property through a reduction process and has a fluorescence quenching property also can make various types of GO based biosensor with aptamer [15, 16, 17]. Among the various aptamer-based GO-sensor studies, the most actively studied is a sensor targeting thrombin [18].

  1. One more point is a reference to PoC devices. The sensors were not tested with blood plasma, and their applicability to real still unclear. My last comment is about the analytical characteristics of the sensor.

(Answer)

We totally agree with what you comment. And the further research will be processed with blood in future. However, in this paper, we utilized three types of sensors according to its transducing method. And we focused on comparison of the sensors with same aptamer and thrombin for aptamer analysis. For the comparison, the real samples are abundant, because of the non-specific and uncertain reaction from the sample. So, we didn’t inform the sensor’s application with real sample.

  1. The linear correlation cannot be provided for the data with R2=0.82.

(Answer)

The group of data was analyzed by linear regression method and its fitting result can be described with this equation; Y=2.516x+1.334. For this, we used the GraphPad Prism 8. In the evaluation of fitting data, we were able to obtain an r-square value of 0.8282, and its Sy.x value was 1.192.

where K is the number of parameters fit by regression. The value n-K is the number of degrees of freedom of the regression.

  1. The SD for the figure 1d is necessary to estimate the reproducibility of the signal.

 (Answer)

Thanks for pointing it out. I modified the data with its standard deviation. Figure 2a is also edited in the same way.

Reviewer 3 Report

Some major points

(A) In p4, between lines 146 and 150 the authors discuss about the nature of biomolecular charge influencing the observations of the electrical method. In Fig. 2B, it is observed that electrical resistance initially decreases after 250 pM thrombin is introduced in the system and then it starts increasing, but it still remains negative compared to when there was no thrombin.

The authors may want to discuss the mechanism behind this. According to my literature review it appears that the thrombin reacts with the surface bound aptamer , according to Langmuir kinetics, and once the reaction saturates (when there is no thrombin left to react with aptamer) the change in resistance stabilizes by 30 mins.

Interestingly, a similar mechanism is observed in the results published by Gosai et al(https://doi.org/10.1016/j.bios.2018.10.010). Their system was different, but the mechanism was driven by the charge effect of thrombin and aptamer. Coincidentally, Fig. 4B of the present manuscript shows that the resistance decreases initially as more thrombin reacts with aptamer and then it starts increasing again. I think the authors may want to use this in their explanation. Also remember that thrombin is positively charged in pH 7.4.

(B) The authors should inform the exact process of attaching the TBA to the cantilever. They inform that when thrombin is injected into the system, it binds with the TBA, and the entire complex undergoes desorption from the surface of the cantilever. This could be due to the non-specific adsorption of aptamer. The authors must clarify this point with some relevant reference. They are encouraged to check the publication of Plaxco group. (https://pubs.acs.org/doi/10.1021/la800801v)

(C) As the authors have designed a sensor with pM level of detection, its important that they compare past studies that used similar or different techniques and achieved pM or nM level of detection. Ultimately, it would be beneficial for the readers to understand the advantages / disadvantages of the author’s work in comparison to previous publications. A comparison table may please be included.

Some e.g. are provided :

(1) de la Escosura-Muñiz A; Chunglok W; Surareungchai W; Merkoçi A, Nanochannels for diagnostic of thrombin-related diseases in human blood. Biosensors and Bioelectronics 2013, 40 (1), 24–31

(2) Xiao Y; Lubin Arica A; Heeger Alan J; Plaxco Kevin W, Label-Free Electronic Detection of Thrombin in Blood Serum by Using an Aptamer-Based Sensor. Angewandte Chemie International Edition 2005, 44 (34), 5456–5459.

(3) Gosai A, Hau Yeah BS, Nilsen-Hamilton M, Shrotriya P. Label free thrombin detection in presence of high concentration of albumin using an aptamer-functionalized nanoporous membrane. Biosens Bioelectron. 2019 Feb 1;126:88-95. doi: 10.1016/j.bios.2018.10.010. Epub 2018 Oct 18. PMID: 30396022; PMCID: PMC6383723.

(4) Zhao X-P; Cao J; Nie X-G; Wang S-S; Wang C; Xia X-H, Label-free monitoring of the thrombin–aptamer recognition reaction using an array of nanochannels coupled with electrochemical detection. Electrochemistry Communications 2017, 81 (Supplement C), 5–9.

(5) Allsop T; Mou C; Neal R; Mariani S; Nagel D; Tombelli S; Poole A; Kalli K; Hine A; Webb DJ; Culverhouse P; Mascini M; Minunni M; Bennion I, Real-time kinetic binding studies at attomolar concentrations in solution phase using a single-stage opto- biosensing platform based upon infrared surface plasmons. Opt. Express 2017, 25 (1), 39–58.

The authors are encouraged to find a few more examples if needed.

Minor points

1.        The pH of the thrombin injected should be mentioned.

2.       The sensitivity, bandwidth and resolution of the OSA should be mentioned.

3.       The number of data points for each of the plots/charts should be mentioned.

4.       Standard deviations should be mentioned in figure captions.

5.       What was the reason behind choosing 1550 nm wavelength in optical method ?

Some recent articles on graphene sensor may be mentioned in the review.

Author Response

Reviewer 3

  1. (A) In p4, between lines 146 and 150 the authors discuss about the nature of biomolecular charge influencing the observations of the electrical method. In Fig. 2B, it is observed that electrical resistance initially decreases after 250 pM thrombin is introduced in the system and then it starts increasing, but it still remains negative compared to when there was no thrombin. The authors may want to discuss the mechanism behind this. According to my literature review it appears that the thrombin reacts with the surface bound aptamer, according to Langmuir kinetics, and once the reaction saturates (when there is no thrombin left to react with aptamer) the change in resistance stabilizes by 30 mins. Interestingly, a similar mechanism is observed in the results published by Gosai et al(https://doi.org/10.1016/j.bios.2018.10.010). Their system was different, but the mechanism was driven by the charge effect of thrombin and aptamer. Coincidentally, Fig. 4B of the present manuscript shows that the resistance decreases initially as more thrombin reacts with aptamer and then it starts increasing again. I think the authors may want to use this in their explanation. Also remember that thrombin is positively charged in pH 7.4.

(Answer)

Thanks for the question

Before getting to the point, I’d like to talk about the principle of graphene based FET sensor. It has a tendency that the electrical current decrease when positively charged target reacts. However, in the case of figure 2B, it has shown that the result after desorption of aptamer-thrombin complex from sensor. Thus, the electrical resistance would be recovered to its initial value, and the increased electrical resistance become lower. As a result, the electrical resistance change ratio become having a negative value.

We concluded from numerous repeated experiments that the longer the reaction time and the thicker the thrombin concentration, the more likely the aptamer-thrombin complex will re-adsorb. The data indicate this, and while the electrical sensor can easily respond to very low concentrations of aptamer, it has a limit in measuring high concentrations of thrombin. That is what I intended to convey in this paper.

  1. (B) The authors should inform the exact process of attaching the TBA to the cantilever. They inform that when thrombin is injected into the system, it binds with the TBA, and the entire complex undergoes desorption from the surface of the cantilever. This could be due to the non-specific adsorption of aptamer. The authors must clarify this point with some relevant reference. They are encouraged to check the publication of Plaxco group. (https://pubs.acs.org/doi/10.1021/la800801v)

(Answer)

We attached aptamers onto cantilever’s surface by physical absorption method. It means that there was no chemical immobilization like in the paper of Plaxco group. In the case of that paper, they have to distinguish between immobilized DNA and absorbed DNA because they introduced the chemical immobilization for tight binding of DNA to electrode’s surface whereas we focused on non-immobilization based sensing methods.

 However, we fully agree with that confuse and we’re sorry to that. So, we edited the contents for better understanding as follows;

[3.2.1.] Detection of thrombin with mechanical method
(261~271 lines)
 [before]
 This result could be deduced as a consequence of desorption of the TBA due to its high affinity with thrombin. The difference between the two signals was calculated by equation (1), and the quantity of the signal dropped due to the aptamer response and was recovered at a yield of 132.831%.
 [after]
 This result was based on a consequence of desorption of the TBA due to the deformation of its structure when it reacts with thrombin. Because the TBA was physically absorbed onto the cantilever’s surface, the deformation of TBA could make it detached from the surface. The following difference between the two signals was calculated by equation (1), and as a result, the quantity of the signal dropped due to the aptamer response and was recovered at a yield of 132.831 %.

  1. (C) As the authors have designed a sensor with pM level of detection, it’s important that they compare past studies that used similar or different techniques and achieved pM or nM level of detection. Ultimately, it would be beneficial for the readers to understand the advantages / disadvantages of the author’s work in comparison to previous publications. A comparison table may please be included.

(Answer)

Method

Substrate

Limit of Detection

Refer.

Electrochemical method

Voltammetric sensor

anodized alumina oxide filter membrane (AAO)

~6.82nM
(1.8 ng/mL) in Whole Blood

[1]

Voltammetric sensor

Au electrode

~20nM in serum (sterile water & Fetal calf serum)

[2]

Impedance sensor

Nanoporous anodized alumina or aluminum oxide (NAAO) membranes

10pM in PBS (pH 7.4)

[3]

Potentiometric sensor

PAA membrane

1pM in PBS (pH 7.4)

[4]

Optical Method

SPR sensor

Optical fiber (Au-SiO2-Ge/Fiber)

50aM in binding buffer(50mM Tris+140mM NaCl+1mM MgCl2)

[5]

Thank you for suggestion. We examined those papers closely and categorized according by simple items; measurement method / substrate for immobilization of aptamer / LOD

However, unlike the papers you proposed, our research is based on the principle of adsorption and desorption, and the substances used as substrate for measurement and immobilization in those papers are not based on graphene, so we decided that it’s difficult to compare them with our research as table. Nevertheless, we can find other research which has analogous point with our study and using it, we think it’s possible to add sentences for clarifying the point of our research with the research you sent. Thus, we’d like to add the following sentences at the end of current conclusion part.

Among aptamer sensors, that use thrombin as a target material is being actively researched [28, 29, 30]. However, there are some limitations in using it as a monitoring method for GO-SELEX since studies using graphene-based nanomaterials among them utilize chemical immobilization methods such as EDC/NHS [31, 32, 33]. Moreover, even if there are papers based on adsorption, it is difficult to find papers using the same material in sensing such as graphene oxide, so it is difficult to assess and select an appropriate sensing method depending on purposes [34, 35]. Therefore, we think that the results of this study are possible to propose a reference point that can be used alone or in combination with a monitoring method or a sensing method in accordance with various purposes of various aptamer and thrombin applications.

  1. The pH of the thrombin injected should be mentioned.

(Answer)

The thrombin used in this paper was based on phosphate buffered saline solution with pH 7.4 and experiments were also used same buffer. This information was added into the part of “2.1. Chemical reagents”, 98-line(3pg).

  1. The sensitivity, bandwidth and resolution of the OSA should be mentioned.

(Answer)

We added the information about the optical spectrum analyzer as follows;

optical signal analyzer (ANDO, AQ6319, bandwidth: 600~1700nm, wavelength accuracy : ±10pm, resolution : 0.02nm, resolution accuracy : ±6%) at 1550 nm wavelength

  1. The number of data points for each of the plots/charts should be mentioned. Standard deviations should be mentioned in figure captions.

(Answer)

We added the missing data points and standard deviation values to the inset graph of figure 1-(d) and figure 2-(c) and modified its captions. We also reflected this to the manuscript.

[Figure 1]

Figure 1. Experimental principle and schematic diagrams are based on each measurement method and experimental results after TBA absorption on the sensing substrate. Figure (a and d) Illustration and a representative graph showing the mechanical method. (b and e) Illustration and representative graph showing the electrical method. (c and f). The standard deviation (SD) value after TBA reaction in figure 1d was about 0.018. all experiments except for the highest concentration of TBA in figure 1e, have SD values under 0.74. The highest concentration condition, 20nM TBA, has a SD value of 2.681. In figure 1f, the SD value of two points under 20nM TBA concentration were 3.081 and 5.493 by ascending order.

[figure 2]

Figure 2. Detection of thrombin via mechanical, electric, and optical methods. (a) Mechanical analysis before and after the addition of thrombin on the aptamer adsorbed substrate. (b) Electrical analysis resistance change rate according to the reaction time of thrombin. (c) Optical analysis for the selectivity of the TBA-FAM/GO-based system. The letters (a-g) of the graph and inset picture represent the blank, thrombin, GOx, IgG, trypsin, lysozyme, and ficin. (d) The corresponding calibration plot to (c) shows the fluorescence intensity versus thrombin concentration. The SD value after thrombin reaction in figure 2a was about 0.152. In figure 2b, the SD values increased over time from 0.020 to 0.038. The coefficient of variation of experimental results in figure 2d has an average value of 0.028 (± 0.016).

  1. What was the reason behind choosing 1550 nm wavelength in optical method?

(Answer)

In our previous study, we design and fabricate the cantilever integrated waveguide for modulator of telecommunications. So, we also utilized the device in this paper. That’s why we use 1550nm wavelength.

“A compact and low-driving-voltage silicon electro-absorption modulator utilizing a Schottky diode operating up to 13.2 GHz in C-band”, Japanese Journal of applied physics, 59, 122001, 2020, (https://doi.org/10.35848/1347-4065/abc39f)

Round 2

Reviewer 2 Report

The authors have improved the manuscript, but several point remain without the answers. Very similar experimental setup have been used previously. I don't feel the scientific novelty is sufficient for further publication

Author Response

We try to reply with attached file. 
